## PERSPECTIVE

### Synaptic loss in motor neurons precipitates age-related dysphagia: Middle agers gotta keep in touch!

Ken D. O'Halloran 

*Department of Physiology, University College Cork, Cork, Ireland*

Email: k.ohalloran@ucc.ie

The peer review history is available in the Supporting Information section of this article (https://doi.org/10.1113/JP288488#support-information-section).

Swallowing is a critical function that must be coordinated with breathing to ensure the appropriate transit of food and fluid through the pharynx to the oesophagus without entry into the airways, which can be catastrophic. A highly sophisticated brainstem network coordinates the complex neuromuscular machinery responsible for swallowing behaviours (Pitts & Iceman, 2023). Impairments in swallowing are common in people with neuromuscular and neurodegenerative disorders. Dysphagia is generally under recognised and underappreciated clinically (beyond specialists) and receives much less attention in pre-clinical neurophysiological studies than breathing, which itself could be considered somewhat niche in the grand scheme of basic and translational science.

Yet, disruptions to swallowing and breathing, which are also common in ageing, dramatically increase the risk of aspiration pneumonia and contribute to increased morbidity and mortality (Almirall et al., 2024). It is imperative that we glean a better understanding of the central circuitry and neuromuscular mechanisms controlling swallowing, and gain knowledge of the temporal features contributing to swallow dysfunction in ageing and disease, so that we might develop better interventions to protect and preserve function to improve quality of life.

In this issue of *The Journal of Physiology*, Fogarty (2025) explored swallowing behaviour in 6-month (young), 18-month (middle-aged) and 24-month (old) aged Fischer 344 rats to determine if age-related dysphagia relates exclusively to frank motor neuron loss in the medulla oblongata, a feature of old age, or whether some features of dysphagia precede overt motor neuron death arising due to loss of connectivity of motor neurons *via* age-dependent aberrant dendritic plasticity. The premise for this line of enquiry is that disruption to the sophisticated timing and coordination of swallowing may be a relatively early and arguably greater contributor to dysphagia than decreased strength and efficacy of neuromechanical coupling per se, at least in the context of generally well-preserved laryngeal and pharyngeal muscle quality and function, acknowledging nonetheless that age-related sarcopenia of upper airway muscles can ultimately be a debilitating factor.

In anaesthetised rats, pharyngeal and thoracic oesophageal pressures were measured using solid state transducers. Swallows were evoked by infusing water boluses at the base of the tongue. Pressures were continuously recorded before, during and after swallows. Positive pharyngeal pressures were assessed as an index of swallow strength. Ventilatory (thoracic) pressures were determined during eupnoea (baseline breathing) and schluckatmung. In addition, post-swallow apnoea durations were determined. Schluckatmung, the 'swallow-breath' is characterised by a sub-atmospheric suction pressure in the oesophagus, which facilitates transit of fluids from the pharynx to the oesophagus, an integral component of swallowing. Apnoea, a pause in the central drive to breathe, was judged to be present when respiratory pauses with twice the normal inter-breath duration were observed. In brainstem sections, motor neuron cell body counts were determined unilaterally in the nucleus ambiguus, with careful delineation of the nucleus based on known landmarks and standard stereological principles. Several measures of dendritic complexity were also expertly determined.

In old rats (~50% survival rate), pharyngeal pressures during bolus-induced swallowing were halved with attendant ~20% loss of motor neurons in the semi-compact loose formation of the nucleus ambiguus, which innervates laryngeal and pharyngeal muscles. There were no differences in these parameters between young and middle-aged rats. Interestingly, the ventilatory pressures assessed in the study were unaffected by age, but previous work has established that peak transdiaphragmatic pressure is decreased in old F344 rats associated with diaphragm sarcopenia and weakness (Khurram et al., 2018). Thus, swallow pressures (and peak inspiratory performance) are reduced in old age associated with motor neuron loss.

Of interest, the number of swallows per bolus substantively decreased, whereas apnoea duration substantively increased in middle-aged and old rats compared to young rats revealing age-related perturbations to swallow and ventilatory control. These findings presented from middle-age despite no loss of nucleus ambiguus motor neurons.

Dendritic regression precedes motor neuron death in neurodegenerative disease (Fogarty, 2019). Fogarty (2025) compared dendritic arborisation of nucleus ambiguus motor neurons in young and middle-aged rats. Dendritic arbour length and surface area was reduced in the semi-compact loose formation (but not compact formation) of the nucleus ambiguus in middle-aged rats. Moreover, several measures of dendritic complexity were again found to differ by age and nucleus region, revealing decreased complexity in the semi-compact loose formation in middle-aged rats compared to young rats particularly in distal dendrites, implicated in the integration of excitatory inputs. Consistent with a suggested loss of excitatory input, and motor neuron distress, spine density was reduced in the dendrites of neurons in middle-aged rats.

Collectively the evidence points to degeneration of nucleus ambiguus motor neurons controlling upper airway muscles from middle-age. Prior to the subsequent presentation of motor neuron death, there is evidently a functional consequence of degenerating motor neurons revealed as deficits in the frequency of swallows and swallow-breathing integration. This change, distinct from the timing and efficacy of the distinct sequential phases of swallowing per se, is nevertheless clinically relevant. One wonders if the observations are indicative of degeneration in additional key sites of the brainstem critical to swallow generation and pattern formation, as well as reciprocal pathways between swallow and breathing rhythm generators (Pitts & Iceman, 2023). A widening of the vista of putative age-related decline of central circuits critical to breathing and swallowing

is clearly warranted. This might reveal that dysfunction emerges in middle-age because of pathology at the level of the rhythm generators/oscillators, a true issue of timing. However, even if other critical brainstem sites are protected in middle-age, dendritic regression in the key node of the nucleus ambiguus means that effective transduction of central excitatory cues via the motor nerves to the upper airway is perturbed, an issue of transmission. Strikingly, in the fullness of time, motor neuron loss compounds the problem further, resulting in impaired capacity to achieve peak swallow pressures (Fogarty, 2025). Overt failure may be related to the perfect storm of aberrant timing, transmission and transduction, and further troubled by age-dependent decline in neuromuscular transmission and muscle strength (Matta et al., 2025). Oh, to be young, and to stay young!

In future work, it will be important to establish that swallow efficacy is disrupted in middle-age by methods such as video fluoroscopic analyses to determine aspiration risk. And a silver bullet is needed to halt dendritic regression to maintain central connectivity, no small task. Fogarty's work (2025) is an important advance providing a model and key observations, which should prompt interest and curiosity among basic and clinician scientists. The work provides striking illustration that for preservation of functional capacity in motor systems in later life, middle agers gotta keep in touch!

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

## Additional information

### Competing interests

None declared.

### Author contributions

K.O.: Conception or design of the work; drafting the work or revising it critically for important intellectual content; final approval of the version to be published; agreement to be accountable for all aspects of the work.

### Funding

None.

### Keywords

ageing, dysphagia, dendritic degeneration, motor neurons, nucleus ambiguus, swallowing

## Supporting information

Additional supporting information can be found online in the Supporting Information section at the end of the HTML view of the article. Supporting information files available:

**Peer Review History**

