## [Peer Review History · The Journal of Physiology]

Synaptic loss in motor neurons precipitates age-related dysphagia: Middle agers gotta keep in touch!

Ken D O'Halloran
DOI: 10.1113/JP288488

Corresponding author(s): Ken O'Halloran (k.ohalloran@ucc.ie)

Review Timeline:

Submission Date:	10-Jan-2025
Accepted:	16-Jan-2025

Senior Editor: Harold Schultz

Reviewing Editor: Harold Schultz

Transaction Report:

Dear Professor O'Halloran,

Re: JP-P-2025-288488 "Synaptic loss in motor neurons precipitates age-related dysphagia: Middle agers gotta keep in touch!" by Ken D O'Halloran

We are pleased to tell you that your paper has been accepted for publication in The Journal of Physiology.

Yours sincerely,

Harold Schultz
Senior Editor
The Journal of Physiology

If you would like to receive our 'Research Roundup', a monthly newsletter highlighting the cutting-edge research published in The Physiological Society's family of journals (The Journal of Physiology, Experimental Physiology, Physiological Reports, The Journal of Nutritional Physiology, and The Journal of Precision Medicine: Health and Disease), please click this link, fill in your name and email address and select 'Research Roundup':

<https://www.physoc.org/journals-and-media/membernews>

- You can help your research get the attention it deserves! Check out Wiley's free Promotion Guide for best-practice recommendations for promoting your work at: www.wileyauthors.com/eeo/guide. You can learn more about Wiley Editing Services which offers professional video, design, and writing services to create shareable video abstracts, infographics, conference posters, lay summaries, and research news stories for your research at: www.wileyauthors.com/eeo/promotion.

The Corresponding Author will receive an email from Wiley with details on how to register or log-in to Wiley Authors Services where you will be able to place an order

EDITOR COMMENTS

Thank you for submission of your perspective article to the Journal of Physiology for the focus article "Dendritic alterations precede age-related dysphagia and nucleus ambiguus motor neuron death". The article has been reviewed by the focus author and found to be acceptable for publication without further consideration. The article is now accepted for publication. Congratulations for an interesting and insightful analysis. Please consider the Journal of Physiology for your future studies.

REFeree COMMENTS

Referee #1:

This perspectives piece gives an accurate impression of the focus paper, and adds in some context that the focus paper lacks (eg, rhythm oscillators, NMJs) - Very good overall.